# Cellular and molecular dynamics in the lungs of neonatal and juvenile mice in response to *E. coli*

Sharon A McGrath-Morrow[1]*, Jarrett Venezia[2], Roland Ndeh[1], Nigel Michki[1], Javier Perez[1], Benjamin David Singer[3], Raffaello Cimbro[4], Mark Soloski[4], Alan L Scott[2]

[1]Children's Hospital of Philadelphia Division of Pulmonary Medicine and Sleep, Philadelphia, United States; [2]W Harry Feinstone Department of Molecular Microbiology and Immunology, Bloomberg School of Public Health, Baltimore, United States; [3]Division of Pulmonary and Critical Care Medicine, Department of Medicine Northwestern, University Feinberg School of Medicine, Chicago, United States; [4]Department of Medicine, Division of Rheumatology, Johns Hopkins University, School of Medicine, Baltimore, United States

**Abstract** Bacterial pneumonia in neonates can cause significant morbidity and mortality when compared to other childhood age groups. To understand the immune mechanisms that underlie these age-related differences, we employed a mouse model of *Escherichia coli* pneumonia to determine the dynamic cellular and molecular differences in immune responsiveness between neonates (PND 3–5) and juveniles (PND 12–18), at 24, 48, and 72 hr. Cytokine gene expression from whole lung extracts was also quantified at these time points, using quantitative RT-PCR. *E. coli* challenge resulted in rapid and significant increases in neutrophils, monocytes, and γδT cells, along with significant decreases in dendritic cells and alveolar macrophages in the lungs of both neonates and juveniles. *E. coli*-challenged juvenile lung had significant increases in interstitial macrophages and recruited monocytes that were not observed in neonatal lungs. Expression of IFNγ-responsive genes was positively correlated with the levels and dynamics of MHCII-expressing innate cells in neonatal and juvenile lungs. Several facets of immune responsiveness in the wild-type neonates were recapitulated in juvenile *MHCII*$^{-/-}$ juveniles. Employing a pre-clinical model of *E. coli* pneumonia, we identified significant differences in the early cellular and molecular dynamics in the lungs that likely contribute to the elevated susceptibility of neonates to bacterial pneumonia and could represent targets for intervention to improve respiratory outcomes and survivability of neonates.

*For correspondence: mcgrathmos@chop.edu

Competing interest: The authors declare that no competing interests exist.

## Editor's evaluation

The authors use a model of neonatal *E. coli* pneumonia to study differences between early neonates ad juvenile animals. They observe increased monocyte-derived macrophage recruitment in juveniles compared to neonates as well as an increase in IFNΓ-related genes. The data are of potential interest and will advance the field.

## Introduction

During early infancy and childhood, protective effector and memory immune responses in the lungs evolve through cumulative exposure to microbial and environmental antigens (*Gollwitzer et al., 2014*; *Kasahara et al., 2012*). Because it takes time to accrue sufficient exposure to microbial antigens,

children in the youngest cohorts have a higher likelihood of infection-induced severe disease, respiratory impairment, and death from pneumonia (*Heron, 2019*; *Troeger et al., 2018*; *Ely and Driscoll, 2019*; *Chan et al., 2015*). As infants age and accumulate antigenic experience, the lung's immune responses mature and establish trained innate immunity as well as resident memory cells that allow for the timely, efficient, and robust response that can provide protection from severe disease.

This period of relative vulnerability to respiratory pathogens corresponds to a time of rapid alveolar growth; with most of this growth occurring during the first 2 years of life (*Thurlbeck, 1982*). Growth in the lung is accompanied by Th2-mediated tissue remodeling (*Roux et al., 2011*; *Torow et al., 2017*) that can be easily disrupted by Th1 and Th17 inflammation induced by microbial challenges. An attenuated pathogen-induced immune response may be important to allow for ongoing lung growth by minimizing cell cycle growth arrest responses and detrimental fibrotic remodeling. As such, the modified pathogen immune responses in infants may be beneficial or harmful depending on degree of virulence of the pathogen challenge (*McGrath-Morrow et al., 2018*; *Martinez, 2016*). While there are reports on the composition and functional status of alveolar macrophages, conventional dendritic cells, invariant natural killer T cells, and CD4 T cells in the neonatal lung at steady state after antigen challenge (*Gollwitzer et al., 2014*; *Kasahara et al., 2012*; *Roux et al., 2011*; *Guilliams et al., 2013*; *Lee et al., 2000*; *Thome et al., 2016*), the cellular dynamics induced by live microbial challenge in the neonatal and juvenile lungs have received limited attention. In this study, we sought to understand age-related differences in the cellular and molecular responses during the acute phase of the response induced by challenge with live *Escherichia coli*, a pathogen commonly associated with human neonatal pneumonia (*Chisti et al., 2009*; *Duraes et al., 2016*; *Green and Kolberg, 2016*; *Wang et al., 2010*). Employing a live-challenge model of *E. coli* pneumonia, we demonstrate that the cellular dynamics of innate cells in the lungs of neonatal animals markedly differs from that of juveniles. Of particular note, the lungs of neonatal mice challenged with *E. coli*, had both attenuated interstitial macrophage and monocyte responses that were associated with diminished MHCII-mediated responsiveness. These results reveal important new age-related differences in pathogen-induced immune responsiveness in the neonatal lungs that have implications for understanding the pathogenesis, management, and treatment of severe pneumonia in neonates.

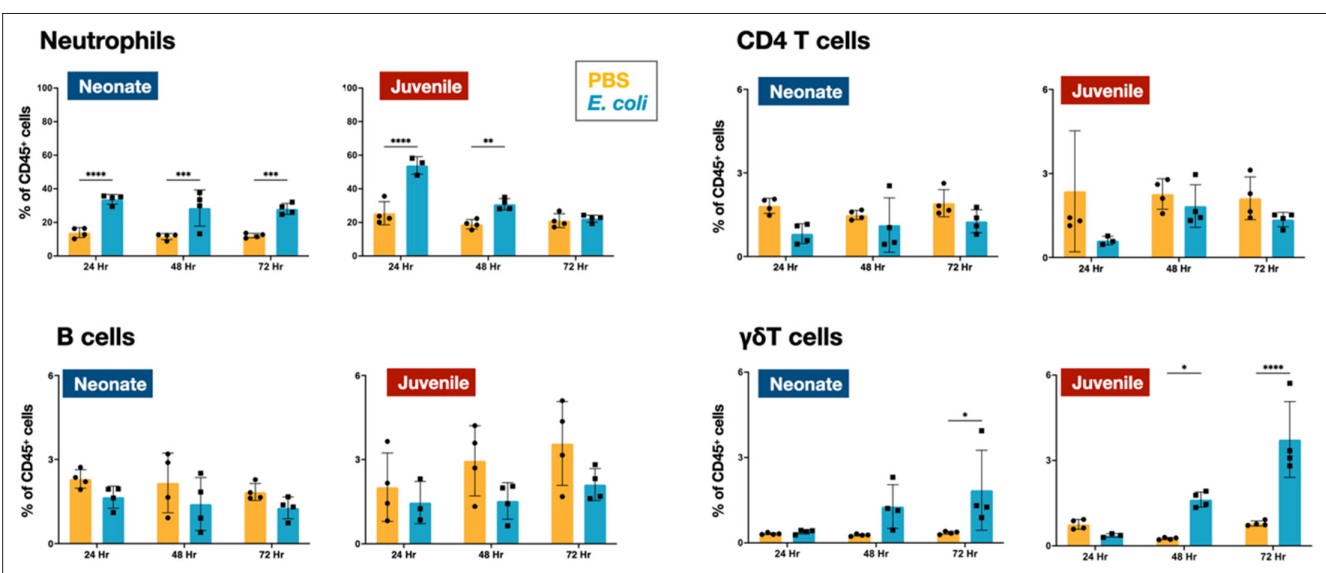

**Figure 1.** *E. coli*-induced changes in neutrophil and lymphocyte dynamics in neonatal and juvenile lungs. Changes in the dynamics of Ly6g[+] neutrophils, CD4[+] T cells, B220[+] B cells, and γδ[+] T cells in neonatal and juvenile lungs at 24, 48, or 72 hr, post-*E. coli* challenge (PEC) as assessed by flow cytometry (see *Figure 3—figure supplement 1* for gating) and expressed as the percentage of total CD45[+] cells. Statistical differences were determined using two-way analysis of variance (ANOVA) with multiple comparisons. Error bars represent standard deviation of the mean. ns – not significant; *p < 0.05; **p < 0.005; ***p = 0.0001; ****p < 0.0001 (*n* = 4 group).

The online version of this article includes the following figure supplement(s) for figure 1:

**Figure supplement 1.** Survival analysis of WT and *MHCII*[−/−] neonatal and juvenile mice after challenge with *E. coli*.

**Figure supplement 2.** Similar bacterial burden between neonatal and juvenile mice at Time$_0$ post-*E. coli* aspiration.

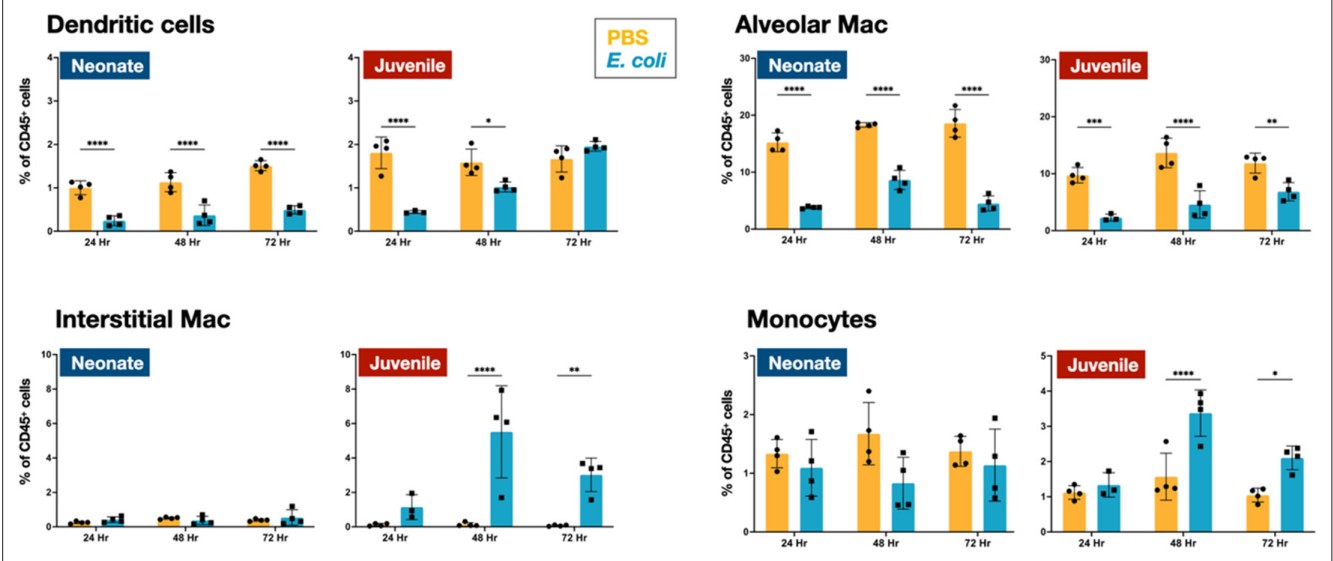

**Figure 2.** *E. coli*-induced changes in mononuclear cell dynamics in neonatal and juvenile lungs. Changes in the dynamics of CD11c+SiglecF−F4/80−MHCIIhi dendritic cells, CD11c+F4/80+SiglecF+ alveolar macrophages, CD11b+F4/80+SiglecF− interstitial macrophages, and CD11b+Ly6c+MHCII+ monocytes in neonatal and juvenile lungs at 24, 48, or 72 hr, post-*E. coli* challenge (PEC) as assessed by flow cytometry (see *Figure 3—figure supplement 1* for gating) and expressed as the percentage of total CD45+ cells. Statistical differences were determined using two-way analysis of variance (ANOVA) with multiple comparisons. Error bars represent standard deviation of the mean. ns – not significant; *p < 0.05; **p < 0.005; ***p = 0.0001; ****p < 0.0001 (*n* = 3 or 4/group).

The online version of this article includes the following figure supplement(s) for figure 2:

**Figure supplement 1.** *E. coli*-induced changes in the number of immune cells in lungs from neonatal mice.

## Results

### Cellular dynamics in neonatal and juvenile lungs in response to *E. coli* challenge

Based on experience with the live *E. coli* challenge model in neonatal and juvenile C57BL/6 mice (*McGrath-Morrow et al., 2017*), the dose chosen to study the cellular and molecular dynamics during the first 72 hr after challenge was 2.4 × 10⁶ CFUs. At this challenge level, we reproducibly achieved >90% survival in both age groups (*Figure 1—figure supplement 1A*), while causing demonstrable inflammation and tissue damage (*Figure 1—figure supplement 1B*).

Flow cytometry was used to define the changes in the CD45+ immune cell populations in the lungs of WT neonatal and juvenile animals at 24, 48, and 72 hr post-*E. coli* challenge (PEC). In neonates, *E. coli* exposure resulted in a rapid increase in neutrophil trafficking to the lungs that persisted through 72 hr PEC (*Figure 1*; *Figure 2—figure supplement 1*). In contrast, while neutrophils also trafficked to the lungs of juvenile mice early, they dropped to control levels by 72 hr PEC. Although the proportions of CD4 T and B cells did not significantly change during the early response in neonates and juveniles, there was a trend in both age groups for an increase in γδT cells at the later time points (*Figure 1*; *Figure 2—figure supplement 1*).

In the lungs of neonates, while live *E. coli* challenge resulted in early and sustained drops in the proportion of dendritic cells (*Figure 2*) and alveolar macrophages (*Figures 2 and 3A*), the numbers of these two mononuclear cell populations were elevated through 72 hr PEC (*Figure 2—figure supplement 1*). The different trends for the cell number and cell proportions are likely due to the large influx of neutrophils into the lungs (*Figure 2—figure supplement 1*). There was also a rapid drop in the proportion of dendritic cells and alveolar macrophages in the lungs of juveniles, but in contrast to neonates, these cells recovered to control levels by 72 hr PEC, albeit with a distinct alteration in surface expression of F4/80 and SiglecF, suggesting an altered activation status (*Figures 2 and 3A*). While the lungs from *E. coli*-challenged juveniles displayed a sustained expansion of the interstitial macrophage compartment, there was no change to the constitutively low levels of this cell population in neonatal lungs (*Figures 2 and 3B*). The dynamic changes in the levels of Ly6c expression on the

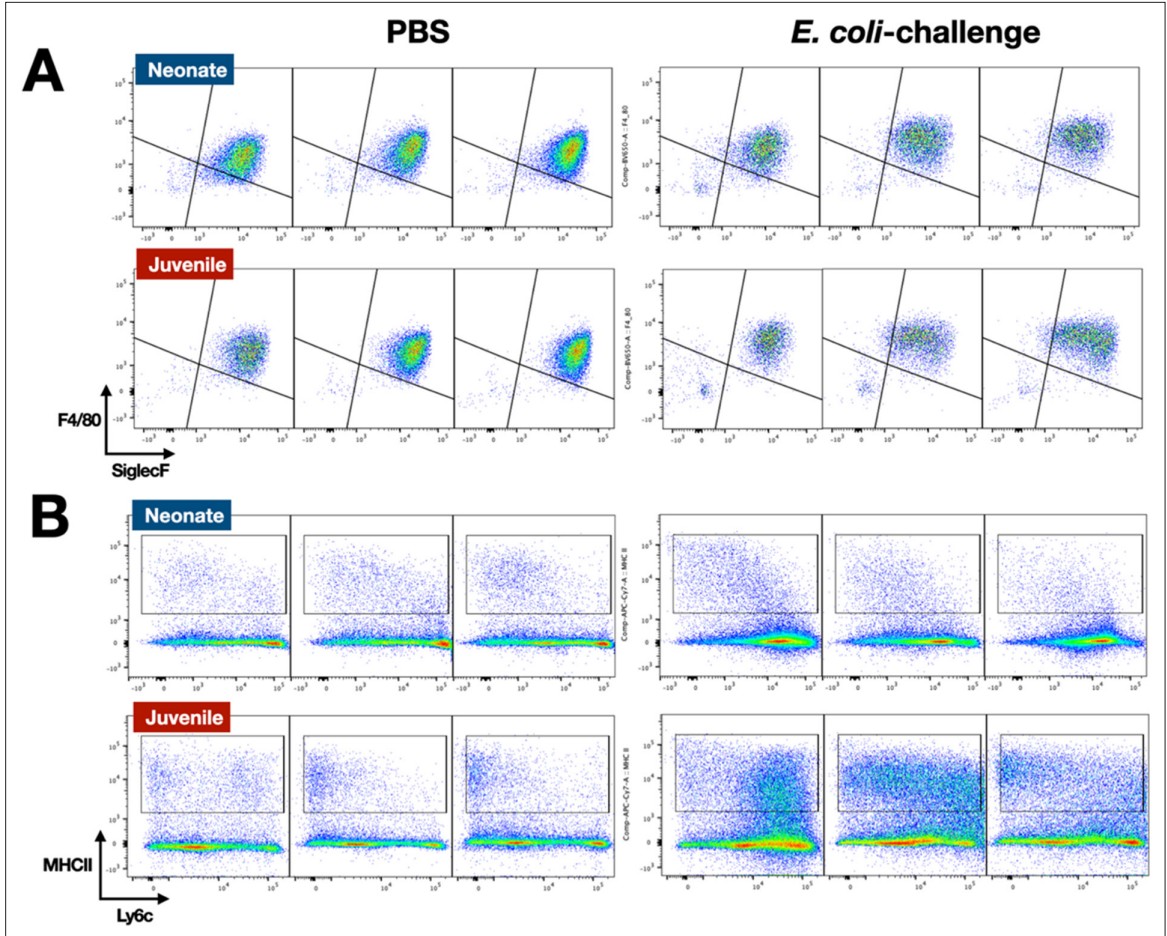

**Figure 3.** *E. coli*-induced changes in the surface phenotypes of alveolar macrophage and interstitial macrophage compartments. (**A**) A representative flow cytometry analysis of F4/80+SiglecF+ alveolar macrophages from *E. coli*-challenged or phosphate-buffered saline (PBS)-treated neonatal and juvenile lungs at 24, 48, and 72 hr (see *Figure 3—figure supplement 1* for gating). (**B**) A representative flow cytometry analysis of the CD11b+MHCII+Ly6c^variable interstitial macrophage compartment from *E. coli*-challenged or PBS-treated neonatal and juvenile lungs at 24, 48, and 72 hr (see *Figure 3—figure supplement 1* for gating).

The online version of this article includes the following figure supplement(s) for figure 3:

**Figure supplement 1.** Gating strategy.

surface of CD45+F4/80+CD11b+SiglecF−MHCII+ cells, a heterogenous population that included interstitial macrophages (*Figure 3B*), suggest that recruited Ly6c^hi monocytes contributed to the expansion of this compartment in juveniles in response to *E. coli* challenge as reported by others (*Chakarov et al., 2019*; *Schyns et al., 2019*). This putative recruited monocyte-to-interstitial macrophage transition after *E. coli* challenge was not observed in the neonatal lungs. Although there was a significant increase in monocytes in the juvenile lungs by 48 hr PEC, the proportion of monocytes in the neonatal lungs did not significantly change (*Figure 2*) despite significant changes in numbers (*Figure 2—figure supplement 1*). This analysis identified differences in granulocyte, lymphoid, and mononuclear cell dynamics between neonates and juveniles that might contribute to the enhanced sensitivity of neonatal animals to live *E. coli* challenge (*McGrath-Morrow et al., 2017*). Of particular note, were the disparate temporal dynamics for dendritic cells, alveolar macrophages, interstitial macrophages, and monocytes, all cells that play key roles in innate immunity, tissue repair, and initiating MHCII-mediated adaptive immune responses.

## Cellular dynamics in the lungs of neonatal and juvenile MHCII⁻/⁻ mice to *E. coli* challenge

The differences in the dynamics of dendritic cells, alveolar macrophages, interstitial macrophages, and monocytes suggest that MHCII-mediated responses might account in part for the heightened morbidity and mortality in neonates. Indeed, previous work demonstrated that blocking MHCII during LPS challenge, resulted in neonatal-like responses in the lungs of juvenile mice (*McGrath-Morrow et al., 2015*). With this in mind, we challenged *MHCII⁻/⁻* animals to gain an insight into the influence of MHCII-mediated cellular and molecular responses in the lungs of neonatal and juvenile animals.

The *MHCII⁻/⁻* neonates and juveniles had 80–85% survival at the 72 hr mark after the challenge with $2.4 \times 10^6$ CFUs (*Figure 1—figure supplement 1C*). In the lungs of *MHCII⁻/⁻* neonates and juvenile there were notable levels of cellular infiltrates observed, in response to *E. coli* challenge (*Figure 1— figure supplement 1D*).

We examined the cellular immune dynamics in the lungs from neonatal and juvenile *MHCII⁻/⁻* mice at 48 hr PEC; a time point chosen to represent the peak of lung inflammation induced by the *E. coli* challenge. The total CD45⁺ cell count significantly increased in both WT neonatal and WT juvenile

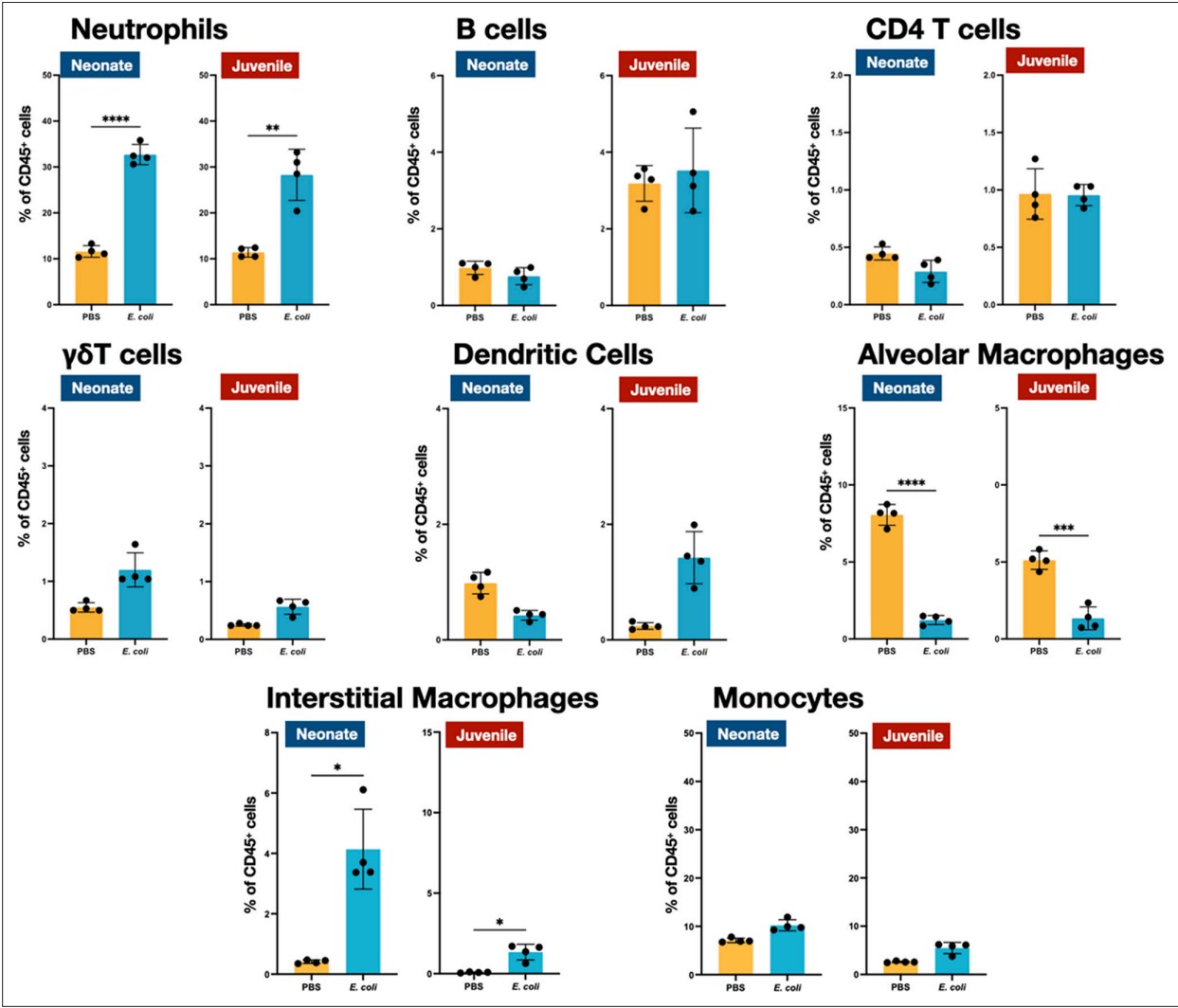

**Figure 4.** *E. coli*-induced changes in the proportions of neutrophils, lymphocytes, and mononuclear cells in the lungs of *MHC⁻/⁻* neonates and juveniles at 48 hr post-challenge with *E. coli*. Changes in neutrophils, CD4 T cells, B cells, γδT cells, dendritic cells, alveolar macrophages, interstitial macrophages, and monocytes in neonatal and juvenile lungs at 48 hr post-*E. coli* challenge (PEC) as assessed by flow cytometry (see *Figure 3—figure supplement 1* for gating) and expressed as the percentage of total CD45⁺ cells. Statistical differences were determined using two-way analysis of variance (ANOVA) with multiple comparisons. Error bars represent standard deviation of the mean. ns – not significant; *p < 0.05; **p < 0.005; ***p = 0.0001; ****p < 0.0001 (n = 3 or 4/group).

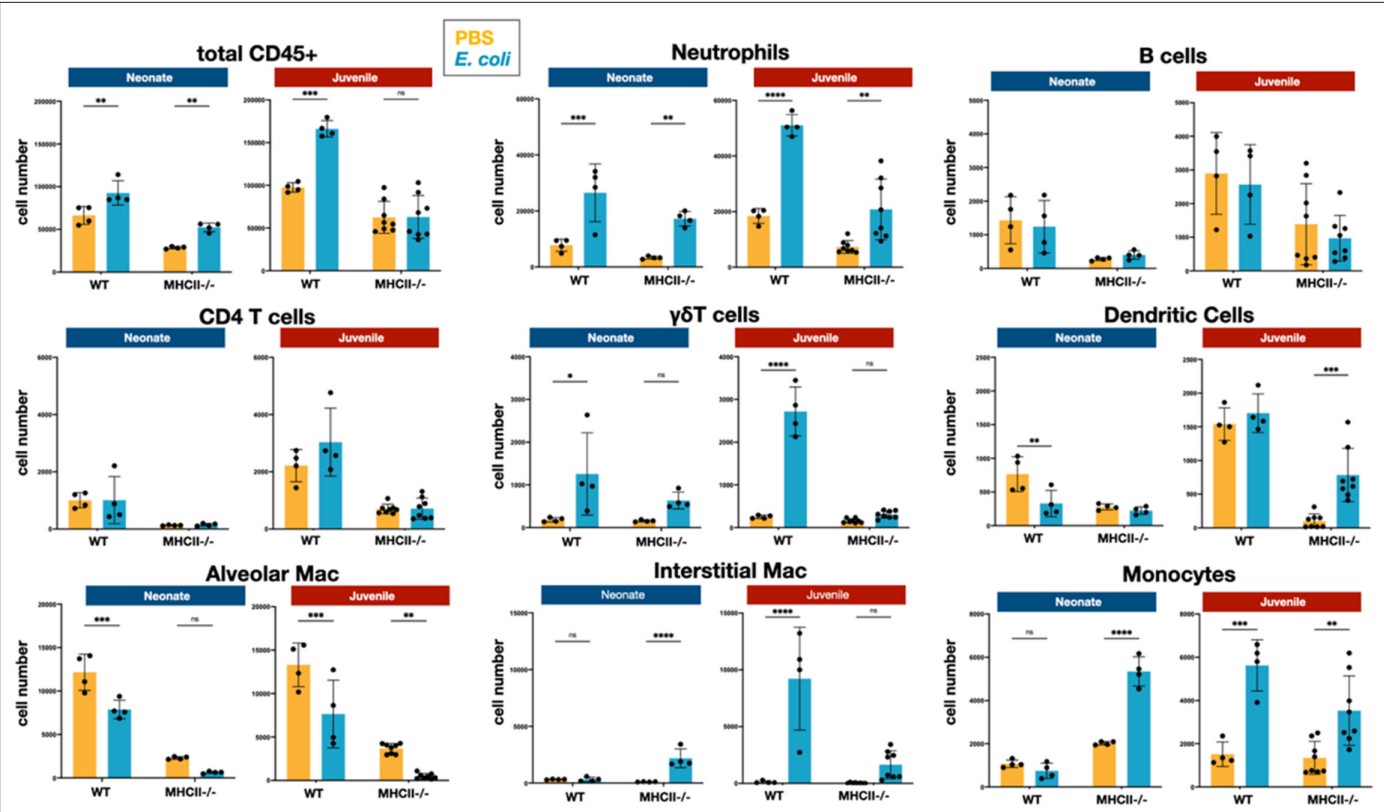

**Figure 5.** *E. coli*-induced changes in the numbers of neutrophils, lymphocytes, and mononuclear cells in the lungs of WT and *MHC$^{-/-}$* neonates and juveniles at 48 hr post-challenge with *E. coli*. Changes in the numbers of total CD45$^+$ cells, neutrophils, CD4 T cells, B cells, γδT cells, dendritic cells, alveolar macrophages, interstitial macrophages, and monocytes in neonatal and juvenile lungs at 48 hr post-*E. coli* challenge (PEC) as assessed by flow cytometry (see *Figure 3—figure supplement 1* for gating). Statistical differences were determined using two-way analysis of variance (ANOVA) with Holm–Šídák post hoc test for multiple comparisons. Error bars represent standard deviation of the mean. ns – not significant; *p < 0.05; **p < 0.005; ***p = 0.0001; ****p < 0.0001 (n = 4–8/group).

lungs in response to *E. coli* challenge, with a substantively larger growth in the juvenile lungs (Figure 5). In contrast, CD45$^+$ cell dynamics was muted in the lungs from *MHCII$^{-/-}$* animals. As observed in WT animals, *MHCII$^{-/-}$* neonates and juveniles had an increase in the percentages and numbers of neutrophils at 48 hr PEC (*Figures 1, 4 and 5*). Although γδT cells showed a proportional increase in the *MHCII$^{-/-}$* lungs (*Figure 4*), this was not reflected in an increase in γδT cell numbers, as seen in the WT animals (*Figure 5*). CD4 T cells were minimal in *MHCII$^{-/-}$* lungs from both age groups. The proportions and numbers of B cells, dendritic cells, and monocytes found in *MHCII$^{-/-}$* neonatal and juvenile lungs at 48 hr PEC were similar to those found in WT animals (*Figures 1, 2, 4 and 5*). Interestingly, compared to the response in WT lungs, the alveolar macrophage response to *E. coli* challenge remained blunted in both *MHCII$^{-/-}$* age groups at 48 hr, PEC. Although the proportions of interstitial macrophages and monocytes were modestly higher in both neonatal and juvenile *MHCII$^{-/-}$* lungs (*Figure 4*), these proportional increases corresponded to significant increases in cell numbers (*Figure 5*). Taken together, these results suggest that MHCII signaling contributes to the early regulation of proliferation and/or activation of key innate cells in the neonatal lung environment.

## Transcriptional response in neonatal and juvenile lungs to *E. coli* challenge

The transcriptional dynamics of select cytokine and chemokine genes were measured from whole tissue extracts to obtain insight into cell activation and trafficking differences between neonatal and juvenile lungs, after exposure to live *E. coli*. The proinflammatory cytokines TNF and IL-6 are induced early downstream of signaling of pattern recognition receptors (*Li and Wu, 2021*). While *tnf* and *Il6* transcription was increased in neonatal and juvenile lungs from both WT and *MHC$^{-/-}$* animals, WT

juvenile lungs had heightened responsiveness at 24 and 48 hr, PEC compared to the other groups (*Figure 6*). As predicted from the enhanced numbers of MHCII-positive cells (*Figure 2*), the WT juvenile lungs had the highest fold increase in *Ciita* expression at 24 and 48 hr, PEC while the lungs from WT neonates and *MHCII⁻/⁻* animals had minimal expression of lung *Ciita* at any time point (*Figure 6*). The *Ciita* transcription pattern supports our prior findings in which we found increased expression of MHCII on bronchoalveolar lavage-derived macrophages from juvenile but not from neonatal mice, challenged with LPS (*McGrath-Morrow et al., 2015*). Studies have also demonstrated that a lack of pathogen-induced *Ciita* expression in the lungs of WT neonatal and *MHCII⁻/⁻* mice is associated with lower numbers of IFNγ-producing CD4 T cells (*van den Elsen et al., 1998*; *Steimle et al., 1994*; *Chang et al., 1994*).

In support of the idea that low *Ciita* expression might be linked to low IFNγ production, the transcription of IFN-inducible genes *Cxcl10* were *Ccl4* were minimal in response to *E. coli* challenge in WT neonatal and *MHCII⁻/⁻* animals (*Figure 6*). However, other IFN-inducible genes – *Ccl2*, *Ccl20*, and *Cxcl1* – showed patterns of expression that suggested that additional factors contribute to the transcriptional regulation of these chemokine genes during the early stages of the innate response to *E. coli* challenge. *Cxcl1*, a Th17-associated chemokine involved in neutrophil migration and NFkβ

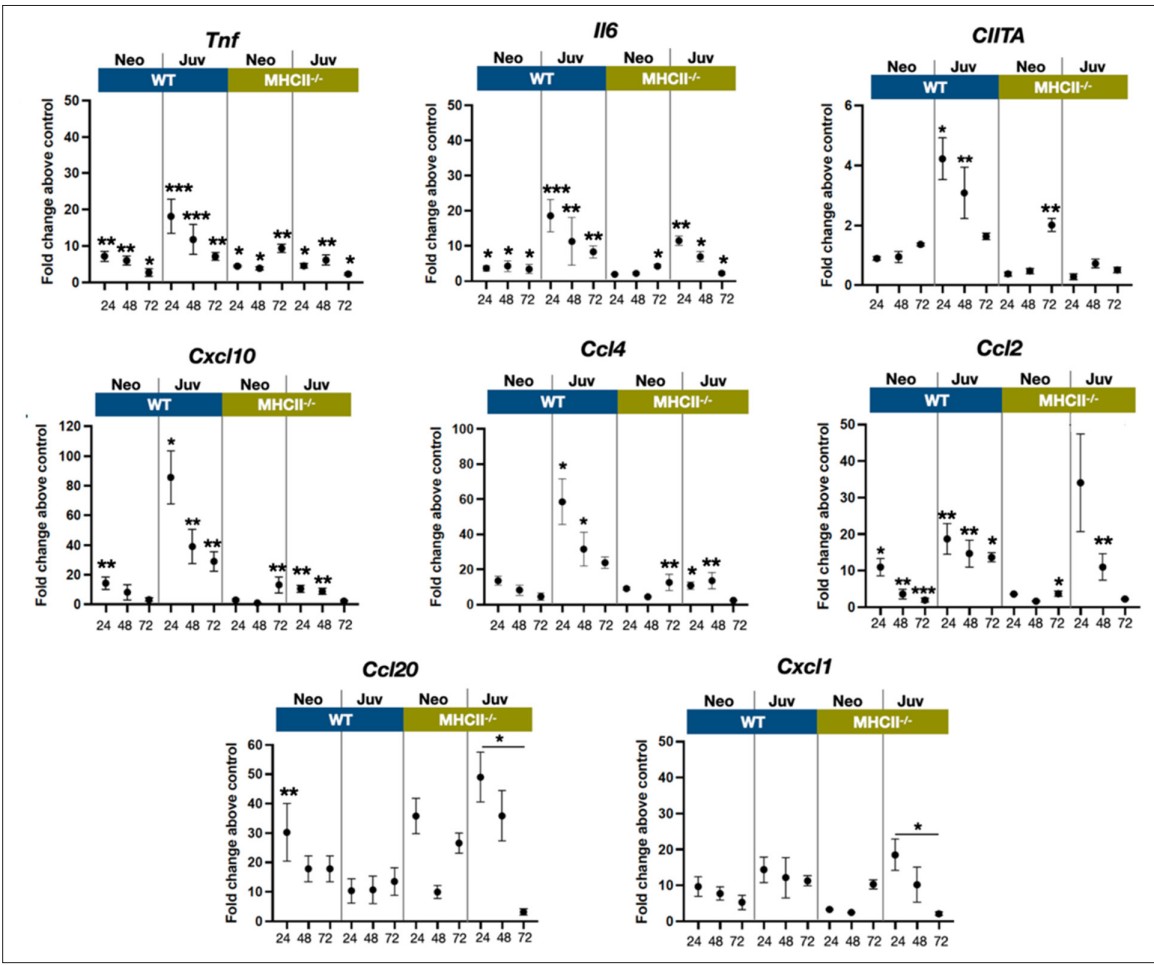

**Figure 6.** *E. coli*-induced changes in the expression dynamics of select regulatory, cytokine, and chemokine genes in WT and MHC⁻/⁻ neonatal and juvenile lungs. Quantitative RT-PCR analysis of total RNA isolated from the lungs of WT and MHCII⁻/⁻ neonates and juveniles at 24, 48, and 72 hr post-*E. coli* challenge for changes in expression of class II major histocompatibility complex, transactivator (*Ciita*), tumor necrosis factor (*Tnf*), interleukin 6 (*Il6*), C-X-C motif chemokine ligand 10 (*Cxcl10*), C-C motif chemokine ligand 4 (*Ccl4*), C-C motif chemokine ligand 2 (*Ccl2*), C-C motif chemokine ligand 20 (*Ccl20*), and C-X-C motif chemokine ligand 1 (*Cxcl1*). Data present as fold change over the value for the corresponding age/genetic background phosphate-buffered saline (PBS) controls. Points represent means of the values from 3 to 5 animals per group (error bars represent standard error of the mean). Statistical significance between groups were determined using one-way analysis of variance (ANOVA) with multiple comparisons. *p < 0.01, **p < 0.004, ***p < 0.001.

activation (*Cai et al., 2010*; *Jin et al., 2014*), and *Ccl2(Mcp1)*, a monocyte chemoattractant induced by the NFkβ pathway (*Bawazeer and Theoharides, 2019*), were also upregulated in *MHCII⁻ᐟ⁻* juvenile lungs at 24 hr PEC (*Figure 6*). Interestingly, the fold changes in *Ccl20* transcription, PEC were substantially lower in WT juvenile lungs compared to WT neonates or *MHCII⁻ᐟ⁻* neonates and juveniles. In addition to its direct antibacterial properties (*Starner et al., 2003*), endothelial cell-derived CCL20 is reported to be chemotactic for CCR6⁺ lymphocytes and dendritic cells and to play key roles in innate and adaptive inflammation at barrier surfaces (*Lee et al., 2013*). The early robust induction of *Ccl20* in WT neonate and *MHCII⁻ᐟ⁻* lungs suggests a role for MHCII-mediated signaling in controlling the level of *Ccl20* transcription.

## Discussion

This study employed a murine model of bacterial pneumonia during infancy to examine age-related differences in the induction of lung immunity. The early responses of neonates and juveniles to live *E. coli* challenge were marked by substantial differences in the temporal dynamics for dendritic and mononuclear cells. Within 24 hr of PEC, there were significant reductions in both dendritic cells and alveolar macrophages that persisted in the neonatal lung but trended toward recovery to control levels by 72 hr PEC in the lungs from juveniles (*Figure 2*). Conventional dendritic cells are a prominent cell population during the first weeks of life in the mouse lungs that exhibit an activation profile that differs from adult-derived lung DCs by being strong inducers of Th2 immune responses (*Gollwitzer et al., 2014*). Colonization by the microbiota induces PD-L1 (programmed cell-death ligand 1) on neonatal lung dendritic cells, which endows them with the capacity to induce regulatory T cells (*Gollwitzer et al., 2014*). It is not clear if the persistent (neonates) or transient (juvenile) reductions observed here for dendritic cells and alveolar macrophages were due to cell death, emigration to local lymph nodes, or activation-associated changes in surface phenotype.

Lung macrophages are comprised of two major populations that occupy distinct anatomical compartments. Alveolar macrophages reside in the bronchoalveolar space and alveoli where these tissue resident cells function in regulating surfactant production, maintenance of epithelial barrier integrity, efferocytosis, and removal of inhaled particles (*Bain and MacDonald, 2022*). In addition, they have traditionally been designated as a first-line defense against air-borne pathogens, however, with the recent appreciation that recruited monocyte-derived macrophages are major contributors to the innate defense against microbial exposure, this aspect of alveolar macrophage function is under assessment (*Aegerter et al., 2020*; *Liao et al., 2020*). The significantly less abundant interstitial macrophages are located in the interstitial spaces between the alveoli and the capillaries as well as in the interstitial environment surrounding conducting airways where they interact with stromal and other innate cells and play roles in inflammation (*Chakarov et al., 2019*; *Schyns et al., 2019*) and immunoregulation (*Ural et al., 2020*).

As noted, there was a significant reduction of alveolar macrophages at 24 hr PEC that persisted in neonatal lungs but was only transient in juvenile lungs (*Figure 2*). Acute loss of tissue resident macrophages at the early stages of inflammation, an occurrence referred to as 'macrophage disappearance reaction', has been described for peritoneal macrophages (*Barth et al., 1995*; *Louwe et al., 2021*) and in the lungs after LPS or CpG DNA challenge (*Bain and MacDonald, 2022*; *Sabatel et al., 2017*). Interestingly, in these studies the transient loss of alveolar macrophages was accompanied by an expansion of interstitial macrophages (*Bain and MacDonald, 2022*; *Sabatel et al., 2017*), similar to that seen in the lungs of WT juveniles challenged with live *E. coli* (*Figures 2 and 5*).

A notable aspect of the cellular dynamics that distinguished the neonatal response from that of the juveniles was the striking differential in the numbers of interstitial macrophages (*Figures 2 and 5*). Current evidence indicates that at steady state, interstitial macrophages are maintained through slow replacement by monocytes that take up residency in the lungs (*Chakarov et al., 2019*; *Schyns et al., 2019*) and that a majority of the increase observed during inflammation is due to monocyte recruitment from the peripheral circulation (*Huang et al., 2018*). Thus, the interstitial macrophage levels are directly connected to the monocyte response, which was more robust in the lungs from juvenile animals. A majority of the increases in interstitial macrophages in juvenile lungs were made up of MHCIIʰⁱ cells (data not shown) which supports the idea that they were derived from recruited monocytes (*Chakarov et al., 2019*; *Schyns et al., 2019*) and could provide a cellular context for the increased *Ciita* expression observed in the whole lungs from juveniles. Given their role in the clearing

of live bacterial challenges (*Pisu et al., 2021*), the apparent lack of interstitial macrophage development in the neonatal lungs could be a contributing factor to the lower ability of neonates to clear a bacteria from the lung in a timely fashion and resolve inflammation (*McGrath-Morrow et al., 2017*).

The attenuated dynamics of MHCII-expressing mononuclear cells observed for neonates in this study likely contributes to the reported differences in MHCII-associated responsiveness between neonates and juveniles. *Lee et al., 2001* found that IFNγ-induced MHCII expression was attenuated in neonatal macrophages. LPS exposure significantly induced expression of MHCII in juvenile but not in neonatal peritoneal macrophages (*McGrath-Morrow et al., 2015*) and aspiration of LPS resulted in increased MHCII on and *Ciita* expression by bronchoalveolar lavage-derived mononuclear cells from juvenile but not from neonatal lungs (*McGrath-Morrow et al., 2015*). This reduced presence of MHCII-expressing cells might also have contributed to the reduced T cell responsiveness and diminished cytokine gene expression after live *E. coli* challenge. In general, there was an elevated transcription of *tnf*, *Il6*, and IFNγ-responsive genes in WT juvenile compared to WT neonatal and *MHCII$^{-/-}$* lungs (*Figure 6*). It is interesting to note that the responses in the lungs from juvenile *MHCII$^{-/-}$* animals at 48 hr PEC resembled the responses induced in the lungs from WT neonatal animals suggesting that restricted MHCII-associated signaling plays a key role in shaping the response in the neonatal lung. Studies carried out at the single-cell level will likely be required to determine the relative contributions of MHCII$^+$ mononuclear cell dynamics, the inherent propensity of CD4 T cells from juvenile mice to express Th1-associated cytokines upon *E. coli* challenge (*McGrath-Morrow et al., 2018*), or other factors in the differential responsiveness in neonatal and juvenile lungs.

Ideas and speculation – It is possible that the attenuated inflammatory response of the neonate has some survival advantages. An overly robust immune response in the neonatal lung could disrupt essential growth dynamics during early postnatal lung development such that blunted responsiveness to pathogens during periods of rapid growth may be important in minimizing disruption to organogenesis while maintaining essential homeostatic and cellular processes. With a better understanding of the innate and adaptive responses in the neonatal lung environment it might be possible to devise interventions that transiently and strategically boost immune responsiveness in the neonate to improve respiratory outcomes to microbial challenge while minimizing the disruption to lung development. The findings reported here underscored differences in the cellular and molecular responses to live *E. coli* challenge that could contribute to the increased morbidity and mortality observed in neonates when challenged with a respiratory pathogen.

## Methods

**Key resources table**

| Reagent type (species) or resource | Designation | Source or reference | Identifiers | Additional information |
|---|---|---|---|---|
| Strain, strain background (*Mus musculus*) | B6;129S2-*H2$^{dlAb1-Ea}$*/J | Jackson Laboratory | Strain #003584 | |
| Antibody | B220/AF488 (Rat monoclonal) | Biolegend | Cat# 103228 | IgG2a (clone RA3-6B2) used at 1/100 |
| Antibody | CD25/PE (Rat polyclonal) | BD | Cat# 561065 | IgG1 used at 1/50 |
| Antibody | CD64/PEcy7 (Mouse monoclonal) | Biolegend | Cat# 139314 | X54-5/7 used at 1/100 |
| Antibody | CD11b/PE-CF594 (Rat monoclonal) | BD | Cat# 562317 | M1/70 IgG2b used at 1/50 |
| Antibody | CD4/BB700 (Rat monoclonal) | BD | Cat# 566408 | RM4-5 IgG2a used at 1/50 |
| Antibody | CD11c/APC-R700 (Hamster monoclonal) | BD | Cat# 565872 | N418 IgG2 used at 1/40 |
| Antibody | PD-1/APC-AF647 (Rat monoclonal) | Biolegend | Cat# 135209 | 29F-1A12 IgG2a used at 1/30 |
| Antibody | MHCII/APC-Cy7 (Rat monoclonal) | Biolegend | Cat# 107627 | M5/114.15.2 IgG2b used at 1/50 |

*Continued on next page*

*Continued*

| Reagent type (species) or resource | Designation | Source or reference | Identifiers | Additional information |
|---|---|---|---|---|
| Antibody | SigF/BV421 (Rat monoclonal) | BD | Cat# 562681 | E50-2440 IgG2a used at 1/50 |
| Antibody | Ly6G/BV510 (Rat monoclonal) | BD | Cat# 740157 | 1A8 IgG2a used at 1/50 |
| Antibody | Ly6C/BV605 (Rat monoclonal) | BD | Cat# 128036 | HK1.4 IgG2c used at 1/100 |
| Antibody | F4/80/BV650 (Rat monoclonal) | BD | Cat# 743282 | T45.2342 IgG2a used at 1/20 |
| Antibody | CD3/BV711 (Hamster monoclonal) | BD | Cat# 563123 | 145-2C11 IgG1 used at 1/40 |
| Antibody | ITCRgd/BV786/SB780 (Hamster monoclonal) | Invitrogen | Cat# 78-5711-82 | GL3 IgG used at 1/30 |
| Antibody | Cd45/BUV395 (Rat monoclonal) | BD | Cat# 564279 | 30-F11 IgG2b used at 1/50 |
| Antibody | CD8/BUV737 (Rat monoclonal) | BD | Cat# 564297 | 5306.7 IgG2a used at 1/40 |

## Mice

Timed pregnant C57BL/6NJ mice were obtained from NCI (Bethesda, MD). Animals were maintained on an AIN 76A diet and water ad libitum and housed at a temperature range of 20–23°C under 12 hr light/dark cycles. All experiments were conducted in accordance with the standards established by the United States Animal Welfare Acts, set forth in NIH guidelines and the Policy and Procedures Manual of the Johns Hopkins University Animal Care and Use Committee (ACUC), protocol # MO16M213. MHCII-knockout mice on a B6 background were obtained from The Jackson Laboratory (B6;129S2-*H2*$^{dlAb1-Ea}$/JStock No:003374 | MHC II$^-$).

## Directly observed aspiration of *E. coli*

Pups were lightly sedated with isoflurane prior to *E. coli* (Seattle 1946, serotype O6, ATCC 25922) aspiration. Mice were given $2.4 \times 10^6$ CFUs of *E. coli* in phosphate-buffered saline (PBS) or PBS alone. Forceps were used to gently extract the tongue, liquid was deposited in the pharynx and aspiration of fluid was directly visualized as previously described (*McGrath-Morrow et al., 2015*). Neonatal mice (PND 3–5) received 10 µl of fluid and juvenile mice (PND 12–18) received 15 µl of fluid.

## Optical density measurements

Neonate and juvenile mice were aspirated with $2.4 \times 10^6$ CFUs of *E. coli* or PBS alone. Mice were then sacrificed immediately post-dosage and their lungs were harvested. Each set of lungs was briefly washed in PBS and then minced using a razor blade. Minced tissue samples were then individually collected into 1 ml of sterile PBS and homogenized using a mechanical rotor homogenizer. A 50 µl droplet of homogenate was then plated onto an LB agar plate, and grown at 30°C for 10 hr, overnight. Remaining homogenate was then used for DNA isolation. The next morning, plates were removed from incubator and droplets were scraped off using a sterile inoculating loop and suspended in 1 ml of sterile LB media. Samples were appropriately diluted to measure the cell density in the sample. This was determined using a spectrophotometer to measure the OD600 (optical density at 600 nM wavelength) for each sample. OD600 absorbance values were then averaged among both age and treatment. *E. coli* treated samples were then normalized to their age-matched vehicle controls and fold difference was measured as an indicator of increased presence of pathogens. OD600 absorbance values were also normalized for lung weight (*Figure 1—figure supplement 2*). No significant differences between OD600 absorbance values were found between the juvenile and neonatal mice aspirated with *E. coli* at Time$_0$, indicating similar bacterial burden between the two age groups.

## Flow cytometry: lung single-cell suspension preparation

The lung was harvested, chopped to small pieces with a razor blade, and suspended in 1 ml of a buffer containing 1.0 mg DNAse 1 (Sigma) and 5.0 mg collagenase II (Worthington Biochem) in 1 ml RPMI. After incubating at 37°C for 30 min, an 18-gauge needle was used to further break up the lung tissue before passing the sample through a 70-μm cell strainer, into a 50-ml conical tube. PBS was added and the sample was centrifuged at 500 × *g* for 5 min. The supernatant was discarded, and 1–2 ml of Ack lysing buffer was added and incubated at room temperature for 5 min. PBS was added to stop the reaction (3× to 5× the volume). The lung sample was then filtered through a 70-μm cell strainer into new 50-ml conical tube and spun down at 500 × *g* for 5 min. FACS buffer was added (3–5 ml depending on the pellet size) and the cells were counted using a Bio-Rad TC20 automated cell counter.

Cells were stained for viability using Live/Dead blue, UV (Invitrogen, L23105) and then with CD16/CD32 (BD Biosciences, 553142) to block Fc receptors. The cells were then stained with the following antibodies/fluorochromes: B220/AF488 (Biolegend, 103228), CD25/PE (BD, 561065), CD64/PEcy7 (Biolegend, 139314), CD11b/PE-CF594 (BD, 562317), CD4/BB700 (BD, 566408), CD11c/APC-R700 (BD, 565872), PD-1/APC-AF647 (Biolegend, 135209), MHC II/APC-Cy7 (Biolegend, 107627), SigF/BV421 (BD, 562681), Ly6G/BV510 (BD, 740157), Ly6C/BV605 (BD 128036), F4/80/BV650 (BD, 743282), CD3/BV711 (BD, 563123), lTCR gd/BV786/SB780 (Invitrogen, 78-5711-82), CD45/BUV395 (BD, 564279), and CD8/BUV737 (BD, 564297). Flow cytometry was performed using a BD Fortessa equipped with five lasers. The gating strategy used to identify lymphoid and myeloid cell populations is outlined in *Figure 3—figure supplement 1*.

## Quantitative RT-PCR

Total RNA was isolated from whole lung at 24, 48, and 72 hr PEC and analyzed as outlined previously (*McGrath-Morrow et al., 2015*). Reverse transcription was performed using total RNA and processed with the SuperScript first-strand synthesis system for RT-PCR according to the manufacturer's protocol (Invitrogen). Quantitative RT-PCR was performed using the Applied Biosystems (Foster City, CA) TaqMan assay system, as previously described (*McGrath-Morrow et al., 2015*). Probes and primers were designed and synthesized by Applied Biosystems. The GADPH gene was used for an internal endogenous control. The following primer–probe sets were used: *Ccl2* (Mm00441242_m1), *Ccl4* (Mm00443111_m1), *Ccl20* (Mm01268754_m1), *Ciita* (Mm00482914_m1), *Cxcl1* (Mm04207460_m1), *Cxcl10* (Mm00445235_m1), *Il6* (Mm00446190_m1), and *Tnfα* (Mm00443258_m1).

## Statistical analysis

Differences in measured variables between treated and control groups were determined using two-way analysis of variance with or without Holm–Šídák post hoc test for multiple comparisons. The number of animals in each group was based on experience and methodological constraints. Comparison of survival curves were determined using log-rank (Mantel–Cox) test. Statistical significance was accepted at $p < 0.05$. Error bars represent standard error of the mean.

## Additional information

### Funding

| Funder | Grant reference number | Author |
| --- | --- | --- |
| National Heart, Lung, and Blood Institute | HL114800 | Sharon A McGrath-Morrow |
| National Heart, Lung, and Blood Institute | HL-140623 | Alan L Scott |

The funders had no role in study design, data collection, and interpretation, or the decision to submit the work for publication.

## Author contributions
Sharon A McGrath-Morrow, Conceptualization, Data curation, Funding acquisition, Investigation, Methodology, Writing - original draft, Project administration, Writing - review and editing; Jarrett Venezia, Visualization, Methodology, Writing - review and editing; Roland Ndeh, Raffaello Cimbro, Methodology, Writing - review and editing; Nigel Michki, Conceptualization, Visualization, Methodology, Writing - review and editing; Javier Perez, Formal analysis, Visualization, Methodology; Benjamin David Singer, Conceptualization, Methodology, Writing - review and editing; Mark Soloski, Formal analysis, Supervision, Methodology, Writing - original draft, Writing - review and editing; Alan L Scott, Formal analysis, Supervision, Writing - original draft

## Author ORCIDs
Sharon A McGrath-Morrow ⓘ http://orcid.org/0000-0002-1576-5394
Jarrett Venezia ⓘ http://orcid.org/0000-0002-8736-0775
Benjamin David Singer ⓘ http://orcid.org/0000-0001-5775-8427
Alan L Scott ⓘ http://orcid.org/0000-0003-0834-728X

## Ethics
All experiments were conducted in accordance with the standards established by the United States Animal Welfare Acts, set forth in NIH guidelines and the Policy and Procedures Manual of the Johns Hopkins University Animal Care and Use Committee (ACUC), protocol # MO16M213.

## Decision letter and Author response
Decision letter https://doi.org/10.7554/eLife.82933.sa1
Author response https://doi.org/10.7554/eLife.82933.sa2

## Additional files

### Supplementary files
• MDAR checklist

### Data availability
Data generated for this study are available through the FlowRepository archive (https://flowrepository.org) under accession numbers FR-FCM-Z63A and FR-FCM-Z63B.

The following datasets were generated:

| Author(s) | Year | Dataset title | Dataset URL | Database and Identifier |
|---|---|---|---|---|
| McGrath-Morrow SA, Scott A | 2023 | Cellular dynamics in the lungs of neonatal and juvenile mice in response to *E. coli* | https://flowrepository.org/id/FR-FCM-Z63A | FlowRepository, FR-FCM-Z63A |
| McGrath-Morrow SA, Scott A | 2023 | Cellular dynamics in the lungs of neonatal and juvenile WT C57BL/6J in response to *E. coli* | https://flowrepository.org/id/FR-FCM-Z63B | FlowRepository, FR-FCM-Z63B |

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
