## [Editor Report]

The authors use a model of neonatal *E. coli* pneumonia to study differences between early neonates ad juvenile animals. They observe increased monocyte-derived macrophage recruitment in juveniles compared to neonates as well as an increase in IFNΓ-related genes. The data are of potential interest and will advance the field.

---

## [Decision Letter]

**Decision letter after peer review:**

Thank you for submitting your article "Cellular and molecular dynamics in the lungs of neonatal and juvenile mice in response to *E. coli*" for consideration by *eLife*. Your article has been reviewed by 2 peer reviewers, including Paul Noble as the Reviewing and Senior Editor and Reviewer #2.

Essential revisions:

Please address the comments provided by reviewer #1

1. The authors would need to provide lung CFU to show they are comparable at both ages as well as the rates of bacteremia (can be assayed by splenic CFU) to make sure the neonatal and juvenile models are truly comparable at the times measurements were made.

2. It is unclear if the lung flow cytometry was performed with the now standard IV anti-CD45 technique developed by the Masopust lab a decade ago to exclude blood contamination. Also the flow cytometry should be detected as both percentages and absolute counts. It is unclear what the authors are calling interstitial macrophages. Many labs are performing IV anti-CD45 and IT anti-CD45 (using a different fluorophore) to define interstitial cells as those that are not labeled with this technique. It appears to the authors are using a set of markers to define these "interstitial" cells and if so that should be clearly stated in the results without the reader having to skip ahead to methods.

3. Given the potential defect in IFNΓ downstream genes, did the authors do ICS to look at IFNΓ+ γ δ T cells or NK cells and are these responses defective?

4. The transcriptional data would need to also be controlled for CFU or at least *E. coli* rRNA to avoid epiphenomena due to higher (or lower) bacterial burdens. Are there differences in MCP1, 3, or 5 that would potentially explain the defect in monocyte/macrophage recruitment.

5. Class II MHC KO mice and humans (bare lymphocyte syndrome) lack CD4^+^ T cells (including Tregs) so the data in the KO would be clearly impacted by this. One could deplete CD4^+^ T cells with GK1.5 to see how much is the lack of CD4^+^ T cells versus

*Reviewer #1 (Recommendations for the authors):*

1. The authors would need to provide lung CFU to show they are comparable at both ages as well as the rates of bacteremia (can be assayed by splenic CFU) to make sure the neonatal and juvenile models are truly comparable at the times measurements were made.

2. It is unclear if the lung flow cytometry was performed with the now standard IV anti-CD45 technique developed by the Masopust lab a decade ago to exclude blood contamination. Also the flow cytometry should be detected as both percentages and absolute counts. It is unclear what the authors are calling interstitial macrophages. Many labs are performing IV anti-CD45 and IT anti-CD45 (using a different fluorophore) to define interstitial cells as those that are not labeled with this technique. It appears to the authors are using a set of markers to define these "interstitial" cells and if so that should be clearly stated in the results without the reader having to skip ahead to methods.

3. Given the potential defect in IFNΓ downstream genes, did the authors do ICS to look at IFNΓ+ γ δ T cells or NK cells and are these responses defective?

4. The transcriptional data would need to also be controlled for CFU or at least *E. coli* rRNA to avoid epiphenomena due to higher (or lower) bacterial burdens. Are there differences in MCP1, 3, or 5 that would potentially explain the defect in monocyte/macrophage recruitment.

5. Class II MHC KO mice and humans (bare lymphocyte syndrome) lack CD4^+^ T cells (including Tregs) so the data in the KO would be clearly impacted by this. One could deplete CD4^+^ T cells with GK1.5 to see how much is the lack of CD4^+^ T cells versus

*Reviewer #2 (Recommendations for the authors):*

This is a very well performed study examining the impact of live *E. coli* infection on the inflammatory and immune response in neonatal and juvenile mouse lungs. The experiments are well performed and the conclusions are supported by the data. The study will inform future studies in this important area of investigation.

---

## [Author Response]

Essential revisions:Please address the comments provided by reviewer #1Comment 1. The authors would need to provide lung CFU to show they are comparable at both ages as well as the rates of bacteremia (can be assayed by splenic CFU) to make sure the neonatal and juvenile models are truly comparable at the times measurements were made and Comment 4. The transcriptional data would need to also be controlled for CFU or at least *E. coli* rRNA to avoid epiphenomena due to higher (or lower) bacterial burdens.

Please see responses below that addresses comments 1 and 3.

We appreciate the reviewer’s concern that CFU aspiration of *E. coli* bacteria into different age groups may elicit different immune responses. The purpose of this study was to identify age-related differences in cellular and molecular dynamics between neonates and juveniles. Multiple factors can affect CFU deposition in the lungs of mice. In this model each mouse was aspirated with either *E. coli* in PBS or PBS alone, while spontaneously breathing under light sedation. By allowing spontaneous breathing in this model, mice were able to aspirate volumes based on the negative pressure generated by a tidal breath, allowing for patchy distribution of infected material throughout the lung, as occurs with a childhood pneumonia. We have previously found age-related immune response differences between the neonate and juvenile, regarding *E. coli* clearance, from the lung. ^1^This observation prompted investigation of the age-related differences in cellular immune cell dynamics between the two childhood age groups. Finally, supplemental figure 1 demonstrates similar histological differences and survival rates between the neonates and juveniles in our model.

Comment 3. It is unclear if the lung flow cytometry was performed with the now standard IV anti-CD45 technique developed by the Masopust lab a decade ago to exclude blood contamination.

It is appreciated that marking immune cells residing within the lung vasculature with labeled anti-CD54 would have facilitated the analysis of the flow cytometry data. However, due to the size of the mice and their veins, performing IV injections of neonatal and juvenile animals are problematic procedures with equally problematic outcomes. Thus, the decision was made not to add intravascular marking to the protocol.

Comment 4: Also, the flow cytometry should be detected as both percentages and absolute counts.

The cell counts for the neonatal WT lungs harvested at 24-, 48-, and 72-hours PEC have been added as Supplemental Figure 3 (see page 28) and comments that refer to these data were added to the Results section on pages 5 and 6 of the revised manuscript. The cell counts for WT and MHCII-/- neonatal and juvenile mice at 48 hours PEC are presented in Figure 5. The data for total cell counts from the WT juvenile lungs at 24-, 48-, and 72-hours PEC are no longer available so total cell counts could not be generated.

Comment 5: It is unclear what the authors are calling interstitial macrophages. Many labs are performing IV anti-CD45 and IT anti-CD45 (using a different fluorophore) to define interstitial cells as those that are not labeled with this technique. It appears to the authors are using a set of markers to define these "interstitial" cells and if so, that should be clearly stated in the results without the reader having to skip ahead to methods.

On page 6, line 115, we define interstitial macrophages as CD45^+^F4/80^+^SiglecF^-^CD11b^+^MHCII+ cells. This set of markers was established based on the surface phenotype of IMs in adult animals at steady state. We concede that the surface phenotype of IM in neonates and juveniles could be different. Further, as implied by the reviewer’s comment, the method for the designation IM is still evolving – especially in the context of lung inflammation. In the Results (page 6) and Discussion (pages 10/11) sections, we acknowledge the established contributions of recruited monocytes to the IM compartment and discuss the IM data from inflamed lungs in a way that concedes that under these dynamic conditions it is difficult to differentiate IM from recruited monocyte.

Comment 6: Are there differences in MCP1, 3, or 5 that would potentially explain the defect in monocyte/macrophage recruitment?

We appreciate Reviewer 1’s comment that differences in Mcp1, 3 or 5 could potentially explain the defect in monocyte/macrophage recruitment. We did find age-related differences between neonates and juveniles with regard to MCP1 (Ccl2) expression which could potentially explain the apparent defect in monocyte/macrophage recruitment.

Figure 6 shows by qRT-PCR analysis, a less robust induction of the monocyte chemoattractant, Mcp1(Ccl2) in WT neonatal lung in response to *E. coli*, compared to WT juveniles. Supporting this finding, we had previously reported that lungs of juvenile mice had significantly higher expression of lung Mcp1(Ccl2) at 48 and 72 h post-*E. coli* compared to neonates by qRT-PCR and that juveniles also had significantly higher levels of Mcp1(Ccl2) protein in the bronchoalveolar lavage at 48 hours post-*E. coli*.^1^ Additional studies examining the relationship between Mcp3 and 5 were not addressed in this study.

Comment 7: Given the potential defect in IFNΓ downstream genes, did the authors do ICS to look at IFNΓ+ γ δ T cells or NK cells and are these responses defective?

We agree with the reviewer that ICS for interferon γ in CD4 and γδT cells would have added to the study. However, in our study design reported here, ICS was not performed due to the low overall yield of both CD4 and γδT cells. In a related study we have employed a single cell approach that will allow us to address this important issue.

Comment 8: Class II MHC KO mice and humans (bare lymphocyte syndrome) lack CD4^+^ T cells (including Tregs) so the data in the KO would be clearly impacted by this. One could deplete CD4^+^ T cells with GK1.5 to see how much the lack of CD4^+^ T cells impacted the response.

In this study, neonatal mice when compared to juveniles, had an attenuated pathogen-induced MHCII driven immune response, associated with a defect in monocyte/macrophage recruitment. In our study we posed the question; would MHCII-/- juvenile mice respond in a similar manner to what was observed in the WT neonatal animals. The results demonstrated that there were indeed similarities between the responses generated in MHCII-/- juvenile and WT neonatal mice implicating a role for MHCII during the early, innate phase of the response to *E. coli* challenge. Of course, this finding is complicated by the fact that the MHCII-/- animals had very low levels of CD4 and γδT cells (see figure 5). While depleting CD4 T cells from WT animals would have been an effective approach to explore the relative contributions of T cells and MHCII to the response in older animals, like IV injections, cell depletion protocols for very young animals are problematic and thus fraught with problems in interpretation. To circumvent these considerable logistical issues, we are exploring alternative approaches to define the role of MHCII in the lungs of neonatal and juvenile mice.

1. McGrath-Morrow, S. A. et al. The innate immune response to lower respiratory tract *E. coli* infection and the role of the CCL2-CCR2 axis in neonatal mice. Cytokine 97, 108-116, doi:S1043-4666(17)30156-4 [pii];10.1016/j.cyto.2017.06.002 [doi] (2017).